# A bioinformatic analysis of the inhibin-betaglycan-endoglin/CD105 network reveals prognostic value in multiple solid tumors

Eduardo Listik[1]☯, Ben Horst[1,2]☯, Alex Seok Choi[1], Nam. Y. Lee[3], Balázs Győrffy[4], Karthikeyan Mythreye[1]*

**1** Department of Pathology, Division of Molecular and Cellular Pathology, University of Alabama at Birmingham, Birmingham, Alabama, United States of America, **2** Department of Chemistry and Biochemistry, University of South Carolina, Columbia, South Carolina, United States of America, **3** Division of Pharmacology, Chemistry and Biochemistry, College of Medicine, University of Arizona, Tucson, Arizona, United States of America, **4** TTK Cancer Biomarker Research Group, Institute of Enzymology, and Semmelweis University Department of Bioinformatics and 2nd Department of Pediatrics, Budapest, Hungary

☯ These authors contributed equally to this work.
* mythreye@uab.edu

**Data Availability Statement:** All relevant data are within the paper and its Supporting information files.

## Abstract

Inhibins and activins are dimeric ligands belonging to the TGFβ superfamily with emergent roles in cancer. Inhibins contain an α-subunit (*INHA*) and a β-subunit (either *INHBA* or *INHBB*), while activins are mainly homodimers of either β$_A$ (*INHBA*) or β$_B$ (*INHBB*) subunits. Inhibins are biomarkers in a subset of cancers and utilize the coreceptors betaglycan (*TGFBR3*) and endoglin (*ENG*) for physiological or pathological outcomes. Given the array of prior reports on inhibin, activin and the coreceptors in cancer, this study aims to provide a comprehensive analysis, assessing their functional prognostic potential in cancer using a bioinformatics approach. We identify cancer cell lines and cancer types most dependent and impacted, which included p53 mutated breast and ovarian cancers and lung adenocarcinomas. Moreover, *INHA* itself was dependent on *TGFBR3* and *ENG/CD105* in multiple cancer types. *INHA*, *INHBA*, *TGFBR3*, and *ENG* also predicted patients' response to anthracycline and taxane therapy in luminal A breast cancers. We also obtained a gene signature model that could accurately classify 96.7% of the cases based on outcomes. Lastly, we cross-compared gene correlations revealing *INHA* dependency to *TGFBR3* or *ENG* influencing different pathways themselves. These results suggest that inhibins are particularly important in a subset of cancers depending on the coreceptor *TGFBR3* and *ENG* and are of substantial prognostic value, thereby warranting further investigation.

## Introduction

Inhibins and activins are dimeric polypeptide members of the TGF-β superfamily, discovered initially as regulators of follicle-stimulating hormone (FSH) [1–9]. Activins are homodimers utilizing different isoforms of the monomeric β$_A$ or β$_B$ subunits located on different

**Funding:** KM, R01CA219495, NCI https://www.
cancer.gov and BG, 2018-2.1.17-TET-KR-00001,
2018-1.3.1-VKE-2018-00032 National Research,
Development and Innovation Office Hungary
(https://ec.europa.eu/growth/tools-databases/
regional-innovation-monitor/organisation/national-
research-development-and-innovation-office). The
funders had no role in study design, data collection
and analysis, decision to publish, or preparation of
the manuscript.

**Competing interests:** NO authors have competing
interests.

chromosomes [10–12]. Inhibin is a heterodimer of an α subunit *(INHA)* and a β subunit
(either β$_A$, *INHBA*, or β$_B$, *INHBB*). Thus the inhibin naming reflects the β subunit in the het-
erodimer: inhibin A (α/β$_A$) and inhibin B (α/β$_B$), respectively [8, 13–16].

Activins, signal primarily through the transcriptional proteins SMAD2/3, much like TGF-β
[17, 18]. Initial receptor binding of activin occurs via type II serine-threonine kinase receptors
(ActRII or ActRIIB). These then recruit and phosphorylate type I serine-threonine kinase
receptors (ActRIB/Alk4 or ActRIC/Alk7) leading to subsequent phosphorylation of SMAD2/3
[8, 17, 19–22]. In multiple tissues, activin signaling is antagonized by inhibin [23]. Thus, the
biological and pathological function of activin is directly impacted by the relative levels of the
mature α subunit. Inhibins, however, have a much lower affinity for the type II receptors com-
pared to activins themselves. The affinity can be greatly enhanced by the presence of the Type
III TGFβ receptor, betaglycan *(TGFBR3)*, which binds inhibin's α subunit with high affinity [8,
19, 23, 24]. Thus the most established mechanism of antagonism by inhibin, is via its ability to
competitively recruit ActRII preventing activin induced downstream signaling in a betagly-
can-dependent manner [8, 19, 23, 24]. This competition model does not allow for direct
inhibin signaling. However, conflicting reports on the presence of a separate high affinity
inhibin receptor [25, 26], recently discovered interactions of the α subunit with the Type I
receptor Alk4 [24], and our recent findings on the requirement of the alternate Type III TGF-
β co-receptor endoglin *(ENG/CD105)* for inhibin responsiveness in endothelial cell function
[27] suggest complex roles for inhibins themselves.

Betaglycan and endoglin, are both coreceptors of the TGF-β superfamily with broad struc-
tural similarities [28–30], including glycosylation in the extracellular domain (ECD), a short
cytoplasmic domain and common intracellular interacting partners [31–36]. Sequence analysis
of betaglycan and endoglin reveal the highest shared homology in the transmembrane (73%)
and cytoplasmic domains (61%), with the most substantial difference being in the ECD
sequence that impacts ligand binding [28–30, 37–39]. Both betaglycan and endoglin knockouts
(KOs) are lethal during embryonic development due to heart and liver defects and defective
vascular development, respectively, highlighting the shared physiological importance of these
coreceptors [40–43]. In contrast to the above-described similarities, betaglycan is more widely
expressed in epithelial cells, while endoglin is predominantly expressed in proliferating endo-
thelial cells [44–46].

In cancer, betaglycan and endoglin impact disease progression by regulating cell migration,
invasion, proliferation, differentiation, and angiogenesis in multiple cancer models [34, 47–
52]. Betaglycan can act as a tumor suppressor in many cancer types and its expression is lost in
several primary cancers [53–55]. However, elevated levels of betaglycan have also been
reported in colon, triple-negative breast cancers and lymphomas, with a role in promoting
bone metastasis in prostate cancer [56], indicating contextual roles for betaglycan in tumor
progression [48, 57, 58]. Endoglin is crucial to angiogenesis, and increased endoglin and
tumor micro-vessel density is correlated with decreased survival in multiple cancers [50, 59].
Evidence in ovarian cancer [60, 61] also suggests that endoglin expression may impact metas-
tasis. Inhibins have been robustly implicated in cancer, and much like other TGF-β members
may have dichotomous, context-dependent effects [62–69]. Inhibins are early tumor suppres-
sors, as the *INHA$^{-/-}$* mice form spontaneous gonadal and granulosa cell tumors [62]. However,
elevated levels of inhibins in multiple cancers are widely reported [63–66, 70, 71]. Active roles
for inhibins in promoting late stage tumorigenesis, in part via effects on angiogenesis, have
also been reported in both prostate cancer [72] and more recently in ovarian cancer [27].

Inhibins have been widely used as a diagnostic marker for a subset of cancers [70, 71, 73]
and both betaglycan and endoglin have been evaluated as therapeutic strategies in cancer.
TRC-105, a monoclonal antibody against endoglin, was tested in twenty-four clinical trials

[74–97]. Current data also suggest benefits of combining anti-endoglin along with checkpoint inhibitors [98]. Similarly a peptide domain of betaglycan called p144 and soluble betaglycan have been tested in multiple cancer types as an anti-TGF-β treatment strategy that decreases tumor growth, angiogenesis, metastasis, and augments immunotherapy [99–106].

Prior and emerging studies reveal the dichotomous functions of inhibin's on cancer depending on either betaglycan [8, 19, 23, 24] or endoglin [27]. Hence, further characterization of the relationship between inhibins-betaglycan-endoglin is vital. This study seeks to provide such prescient information by evaluating the significance, impact, and predictive value of this specific network (*INHA*, *INHBA*, *INHBB*, *TGFBR3*, and *ENG*) by utilizing publicly available genomic and transcriptomic databases.

## Materials and methods

### Public databases data mining

Clinical data, gene expression alterations, and normalized expression data of RNA-seq were obtained from cBioPortal [107, 108]. All available studies were assessed for copy number alterations (CNA) and a subset of cancer for mRNA data (Breast Invasive Carcinoma, Glioblastoma, Lower-grade glioma, Cervical Squamous Cell Carcinoma, Stomach Adenocarcinoma, Head and Neck Squamous Cell Carcinoma, Kidney Renal Clear Cell and Renal Papillary Cell Carcinomas, Liver Hepatocellular Carcinoma, Lung Adenocarcinoma, Ovarian Serous Cystadenocarcinoma, Prostate Adenocarcinoma, Uterine Corpus Endometrial Carcinoma). The results shown here are partly based upon data generated by the TCGA Research Network: https://www.cancer.gov/tcga. Survival data was generated from either KM Plotter [109] or cBioportal (i.e., brain cancers). KM Plotter data for breast, ovarian, lung, and gastric cancer the survival analysis was derived using available gene chip data sets. All others were derived using the RNA-Seq Pan-cancer data sets. The Affymetrix Probe IDs used in gene chip analysis in KM Plotter were: *INHA* (210141_s_at), *INHBA* (204926_at), *INHBB* (205258_at), *TGFBR3* (204731_at), and *ENG* (201808_s_at). Brain cancer data was generated from TCGA Pan Cancer Atlas 2018 dataset for glioblastoma and low-grade glioma. Overall survival (OS) was assessed for all cancer types except ovarian cancer (progression-free survival, PFS) and breast cancer (relapse-free survival, RFS). Gene expression was split into high and low using the median expression. Log-rank statistics were used to calculate the p-value and Hazard ratio (HR).

### Analysis of gene predictiveness to pharmacological treatment

Gene predictive information on treatment regiments was obtained from ROC Plotter (http://www.rocplot.org/) [110]. Gene expression for the analyzed genes was compared using the Mann-Whitney U test. Receiver Operating Characteristic (ROC) plots and significance was also computed. ROC curves were compared using Area Under the Curve (AUC) values, and values above 0.6 with a significant p value were considered acceptable [110]. ROC plot assessment was performed in all pre-established categories in ROC Plotter (i.e., breast and ovarian cancers, and glioblastoma). In breast cancer, subtypes (i.e., luminal A, luminal B, triple-negative, HER2+) were also analyzed separately. Genes of interest were analyzed for complete pathological response in different pharmacological treatments. All available treatment options were investigated including, taxane, anthracycline, platin and temozolomide. Outliers were set to be removed in this analysis and only genes with a false discovery rate (FDR) below 5% were considered.

## Clustering strategies for genes signatures

From the normalized expression data from RNA-seq studies, the Spearman's ρ coefficient was obtained for *INHA*, *TGFBR3*, and *ENG*. These data were clustered through a Euclidean clustering algorithm using Perseus 1.6.5.0 (MaxQuant). Clusters containing high and low correlations sets were isolated and compared in a pair-wise fashion. The derived genes obtained were checked for protein interaction in BioGRID (thebiogrid.org) [111], and later included in pathway analysis, as described in section 2.5. All plots were performed in GraphPad Prism 8.0.

## Gene signature modeling for prognostics

Gene signature modeling was performed using binary probit regression for each set of cancer types related to *INHA*, *TGFBR3*, *ENG* (S5 Table), and their respective outcomes (i.e., positive, 1; or negative, 0). The regression was iterated for presenting only significant elements in the following model:

$$\Pr(Y = 1 | x_1, \ldots, x_k) = \Phi(\beta_0 + \sum_{i=1}^{k} \beta_i x_i)$$

in which $x_i$ are RNA-seq V2 RSEM expression data for each gene, $\beta_i$ are obtained coefficients from this regression, $\Phi$ is the cumulative normal distribution function. Probability values closer to 1 indicate a positive outcome, while values close to 0, indicate a negative outcome. Postestimation of specificity and sensitivity was also implemented. All regression studies were performed in Stata/SE 16.0.

## Pathway assessment

For pathway analysis, DAVID Bioinformatics Resources 6.8 was used to acquire compiled data from the KEGG Pathway Database [112]. Genes for the analysis were annotated to map to human pathways. The significant outputs were then assessed for the percentage of genes from analyzed sets and their relevance. To compare pathways between two sets, a pathway significance value ratio (-$\log_{10}$R), in which R is the ratio, was analyzed. Only pathways with an FDR value below 5% were considered.

## Gene dependency analysis

Gene dependency of *INHA*, *TGFBR3*, *INHBA*, *INHBB*, and *ENG* was analyzed using the DepMap portal (www.depmap.org) [113]. Gene expression from Expression Public20Q1 (accessed between March and April 2020) were compared to the cell line database from CRISPR (Avana) Public20Q1 and Combined RNAi (Broad, Novartis, Marcotte). Gene effect values of less than or equal to -0.5 were used to select dependent genes.

To analyze gene co-dependency, Expression Public20Q1 was compared to all CRISPR and RNAi databases. A gene was considered dependent when correlations between datasets displayed similar trends. Each dependent gene-set was compared between *INHA*, *TGFBR3*, *INHBA*, *INHBB*, and *ENG* to count duplicates. The number of dependent genes were plotted as a Venn diagram.

# Results

## Inhibins and activins are altered in human cancer

We and others reported previously diverse roles for members of the inhibin/activin family in cancer [8, 27, 114–117]. Our and prior mechanistic studies in cancer indicate a strong

dependency of inhibin function on betaglycan and endoglin [24, 27, 118–121]. To begin to evaluate the impact of this relationship more broadly in cancers we analyzed gene alterations including mutations, amplifications, and deletions for the genes encoding inhibin/activin subunits (Fig 1a) *INHA*, *INHBA*, *INHBB*, and the key coreceptors—*TGFBR3*, and *ENG* in all public datasets available in cBioPortal (Fig 1b, S1 Table). While INHBC and INHBE are activin subunits, these were excluded from the analysis as they have not been demonstrated to form heterodimers with INHA [122].

Percentage of patients from the whole cohort that possessed any of the alterations either by themselves or concomitantly was analyzed. We find that melanoma (16.26%), endometrial (13.16%), esophagogastric (10.85%), and lung (10.69%) cancers revealed the highest alterations for the genes. The alterations for the genes varied, with *INHBA* and *TGFBR3* exhibiting higher rates of alterations (0–5.65% and 0.17–3.91% respectively) that also varied by cancer type. The range for *INHA*, *INHBB*, and *ENG* was found to be between 0–2.38%, 0–2.62%, and 0–3.23% respectively (Fig 1b).

In comparing expression levels of each of the genes in the same TCGA datasets as in Fig 1b, we find that overall *ENG* is the most highly expressed gene (Fig 1c) with variance among different cancer types (e.g., lower-grade glioma and cervical *vs.* renal clear cell and lung adenocarcinoma, p < 0.0001) and subtypes (e.g., luminal A *vs.* luminal B breast cancers, p < 0.0001). Interestingly, *TGFBR3* expression differed most notably between glioblastoma and lower-grade gliomas (p < 0.0001). Breast cancers exhibited higher expression as compared to ovarian and endometrial (p < 0.0001) cancers. *INHBB* in contrast was mostly expressed in renal clear cell and hepatocellular carcinomas, which differs from renal papillary cell carcinoma and cervical cancer (p < 0.0001). Both *INHBA* and *INHA* were the least expressed as compared to the others (Fig 1c). Exceptions were head and neck and esophagogastric cancers that had high expression of *INHBA* and lung adenocarcinoma and renal clear cell carcinoma that had high expression of *INHA*.

While the above analysis examined patient tumors, we next examined cell lines as a way to delineate model systems for future studies. For these analyses, we used the DepMap project (www.depmap.org) [113] which is a comprehensive library of human genes that have been either knocked down or knocked out through CRISPR technology in 1,776 human cell lines representing multiple cancer types [123–125]. Dependency scores representing the probability of queried gene dependency for each cell line and thereby cancer type is obtained [126]. Here, we find that the ligand encoding gene *INHA* displayed higher dependency than the activin subunit isoforms *INHBA* or *INHBB* or either receptors *TGFBR3* or *ENG* (Fig 1d). Notably, esophageal, gastric, and ovarian cancers had the highest dependency results for *INHA* ($\geq$ 14%) consistent with the alterations seen in Fig 1c. Within these cancers, *INHBA* exhibited higher dependency values in ovarian cancer (6%) albeit not as high as *INHA*. Besides *INHBB* in myeloma (6%), no other notable dependency relationships were observed.

In an attempt to identify genes most impacted by alteration to each of the individual genes, we examined how RNAi and CRISPR interventions would affect their correlation to specific genes. Those similarly affected by these techniques were found to be dependent on the investigated set of genes. We find that *ENG* exhibited the highest number of dependent genes (Fig 1e, n = 71) followed by *INHBA* (57), *INHBB* (49), *TGFBR3* (44) and *INHA* (30) (Fig 1e, S1 Table). Interestingly, only a total of 5 genes were commonly dependent between *INHA* and the other genes (Fig 1e, *MAX* with *INHBA* and *GRPEL1*, *SF3B4*, *ESR1*, and *TFAP2C* with *INHBB)*. *INHBA* on the other hand had several common dependent genes most notably 13 genes were common with *ENG* dependency (e.g., *VCL*, *TLN1*, and *LYPD3*).

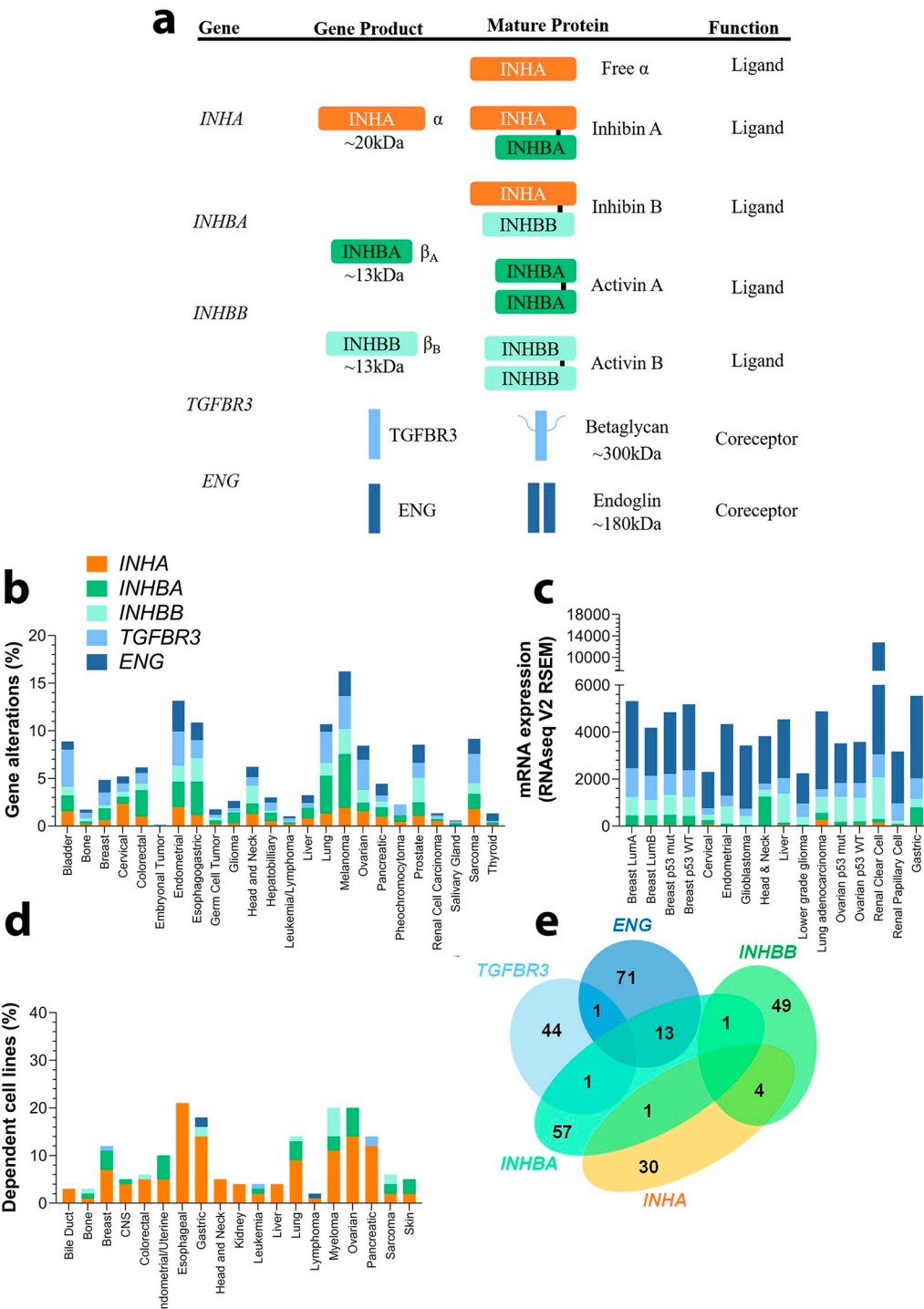

**Fig 1. Expression and gene alterations of inhibin and activins.** (a) Genes encoding INHA, INHBA, and INHBA produce monomeric α and β subunits. These subunits combine to form either homo or heterodimers representing mature inhibin A, inhibin B, activin A, and activin B. (b) TCGA base analysis of gene alteration frequencies of *INHA*, *INHBA*, *INHBB*, *TGBFR3*, and *ENG*. (c) Analysis of gene expression levels, also from TCGA sets, of the same genes as in (b) in a subset of cancer types and subtypes. Analysis included 16 studies and 6258 samples. (d) DepMap analysis of cell line dependency from indicated cancer types on each of the genes in (b). (e) Venn diagram illustrating the number of common dependent genes for each gene in (b). All numeric data are available in S1 Table. Abbreviations—CNS: Central Nervous System; LumA: Luminal A; LumB: Luminal B; mut: mutated; WT: wild-type.

## Effect of inhibins and the coreceptors on patient survival varies by cancer type

Since alterations in expression of inhibin, activin, *TGFBR3* and *ENG* exist in human cancers and prior studies have implicated each of these in patient outcomes [27, 52, 59, 71, 114, 127–130]; we conducted a comprehensive analysis of each of these genes on overall survival (OS), progression-free survival (PFS), or relapse free survival (RFS) in a broad panel of cancers. The goal here was to identify the patients and cancer types most impacted by changes in gene expression. Analysis was conducted using datasets in KM Plotter (Figs 2 and 3, summarized in S2 Table) [109]. For ovarian cancer data sets, only p53 mutated ovarian cancers were included. Patients in KM plotter with p53 mutation status known showed 83% were mutated, cBioportal data sets showed 82.5% frequency of p53 mutation, and it has been reported that over 90% of ovarian cancers present p53 mutations. We find that not all cancers are equally impacted. Of note, we find that in both breast and ovarian cancers all five genes were either positive predictors of survival or non-predictive except *INHBB* in breast (HR = 1.06, p = 0.034) and *INHBA* in ovarian (HR = 1.16, p = 0.047) (Fig 2). However, in p53 mutated cancers, *INHA* was a strong negative predictor of survival for both breast and ovarian cancers (HR = 1.99, p = 0.0056 and HR = 1.55, p = 0.0039, respectively), along with *ENG* in ovarian cancer (HR = 1.36, p = 0.0098, Figs 2 and 3). Additionally, in lung cancers, *INHA* and *ENG* differed from *TGFBR3*, as *INHA* (HR = 1.26, p = 0.00029) and *ENG* (HR = 1.20, p = 0.0056) were both negative predictors of survival while *TGFBR3* (HR = 0.65, p = 3.4E-7) was a strong positive predictor of survival (Fig 2). Specifically, we find that *INHA* and *ENG* are robust predictors of poor survival in lung adenocarcinomas but not in squamous cell carcinomas (Figs 2 and 3). Gastric cancers represent another robust cancer type where all five genes were negatively correlated with survival (Figs 2 and 3). Since *HER2* expression is a frequent abnormality in gastric cancer [131], we examined if there were any differences in survival associated with *HER2* expression. All five genes in both *HER2*[+] and *HER2*[-] gastric cancers, except *INHBA* in *HER2*[-] gastric cancers, were negatively correlated with survival (Fig 2). In kidney cancers, *INHA* was a negative predictor of survival in both renal clear cell and renal papillary cell carcinoma (Figs 2 and 3), consistent with prior findings [27]. *TGFBR3* was a strong positive predictor of survival in both renal clear cell carcinoma (HR = 0.46, p = 2.1E-7) and renal papillary cell carcinoma (HR = 0.53, p = 0.042, Figs 2 and 3). *ENG* (HR = 0.51, p = 8.6E-6) was a positive predictor of survival in renal clear cell carcinoma but not significantly associated with survival in renal papillary cell carcinoma (Fig 2). Finally, in brain cancers, *INHA* was a negative predictor of survival in glioblastoma but a positive predictor in low-grade gliomas (Fig 2). Of note, *ENG* appeared to have a lower range of HR values compared to *INHA* and *TGFBR3*. *INHBA* and *INBBB* were not as significantly correlated with survival as *INHA*, *ENG*, and *TGFBR3*. *INHBA* was significantly correlated with 8 cancer types while *INHBB* was significantly correlated with 9. *INHBA* and *INHBB* showed similar correlations with survival in gastric cancers, specifically *HER2*[+], and renal papillary cell carcinoma (Fig 2). *INHBA* and *INHBB* showed opposing effects however in liver cancer where *INHBA* (HR = 0.62, p = 0.0086) was a strong positive predictor but *INHBB* (HR = 1.52, p = 0.025) was a potent negative predictor (Fig 2).

Since inhibin's biological functions have been shown to be dependent on the coreceptors *TGFBR3* and *ENG* [24, 27, 118–121], we examined the impact of *INHA* based on the expression levels of each of the co-receptor (Table 1). In this analysis, we find that that when separating patients into high or low expressing *TGFBR3* or *ENG* groups (Table 1) in p53 mutated breast cancers, where *INHA* is a negative predictor of survival in all patients (Fig 2), *INHA* was only a predictor of poor survival in patients with low *TGFBR3* (HR = 2.29, p = 0.015) or low *ENG* (HR = 2.24, p = 0.035). Interestingly, this trend was also repeated in renal clear cell

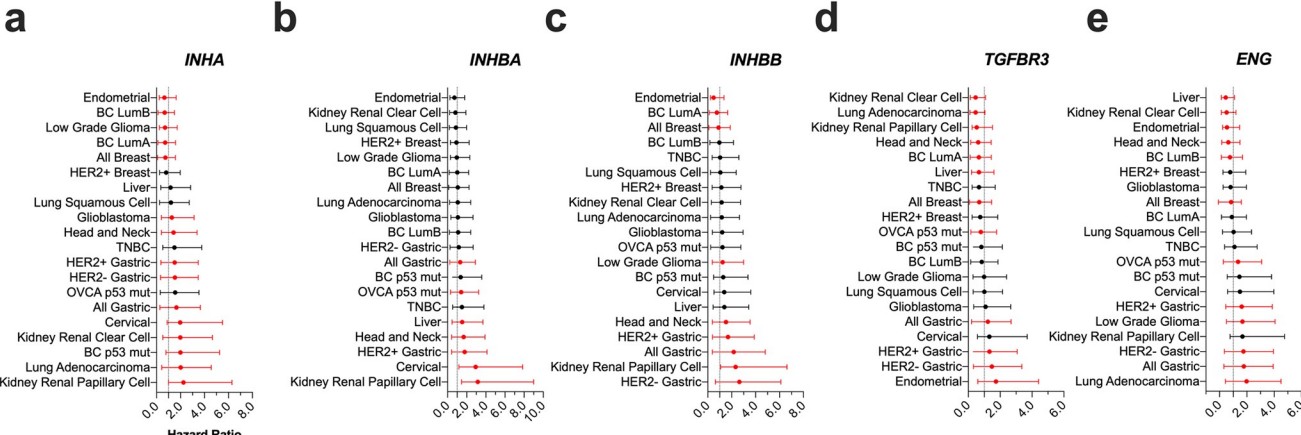

**Fig 2. Impact of *INHA, INHBA, INHBB, TGFBR3,* and *ENG* on patient survival in indicated cancers.** (a) Forest Plot with Hazard Ratios (HR) of indicated genes generated from KM Plotter or data from cBioportal. Black dots represent HR that are not statistically significant (p > 0.05) and red dots represent HR that are statistically significant (p < 0.05). All numeric data are available in S2 Table.

carcinoma, where *INHA* was only a predictor of survival in *TGFBR3* low (HR = 2.75, p = 9.0E-06) and *ENG* low (HR = 2.6, p = 2.5E-06, Table 1). In contrast to breast and renal clear cell cancers where *TGFBR3* and *ENG* both impacted the effect of *INHA* on survival, *TGFBR3* levels did not change *INHA*'s impact on p53 mutated serous ovarian cancers (Table 1). In *ENG* high p53 mutated serous ovarian cancer patients, *INHA* had a more significant negative prediction outcome (HR = 2.12, p = 1.8E-6) compared to *ENG* low (HR = 0.8, p = 0.18, Table 1). Similar outcomes were observed in lung adenocarcinomas with respect to *TGFBR3*, where *INHA* remained a strong negative predictor of survival in patients regardless of *TGFBR3* expression levels (Table 1). However, *INHA* remained a robust negative predictor of survival in lung

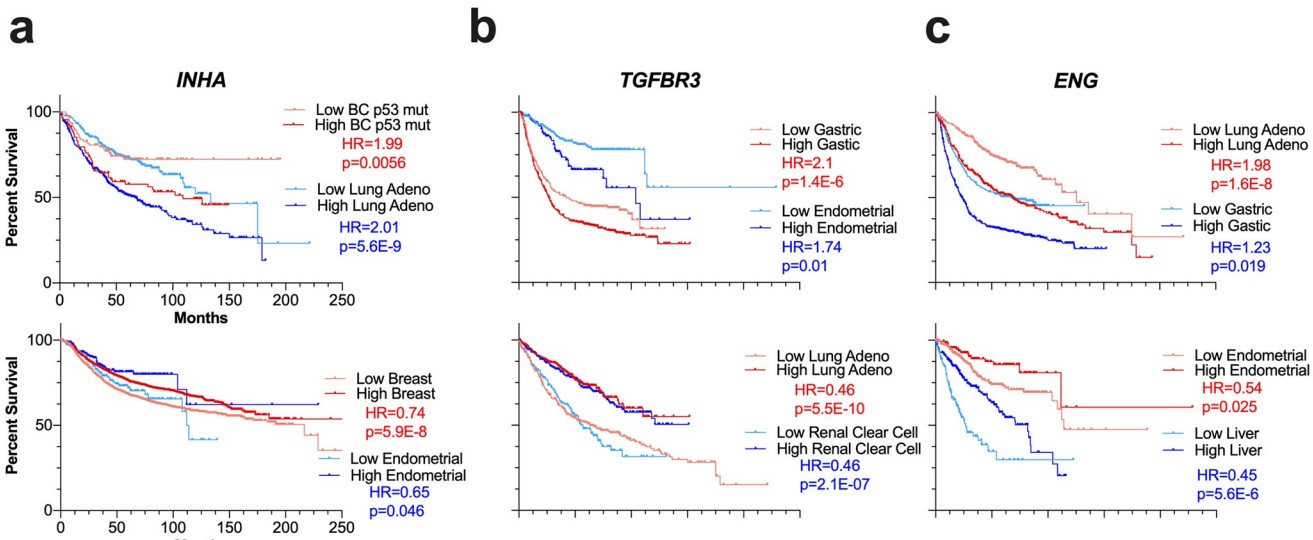

**Fig 3. Representative Kaplan Meier curves for *INHA, TGFBR3,* and *ENG*.** Event-free survival in indicated cancers using median to separate expression (lighter shade indicates bottom patients expressing bottom 50% and darker shade top 50%). Survival curves represent OS for all cancers except breast cancer (RFS) and ovarian cancer (PFS). Top plots show cancer types where the gene is a negative predictor of survival, and bottom plots show cancer types where the gene is a negative predictor.

**Table 1. p values and Hazard Ratios (HR) from survival curves assessing the relationship between *TGFBR3/ENG* and *INHA* on patient survival.**

| Type | Subtype | Variable | *INHA* | *TGFBR3* | *INHA and TGFBR3* | *INHA in High TGFBR3 Patients* | *INHA in Low TGFBR3 Patients* |
|---|---|---|---|---|---|---|---|
| Breast[#] | All | p value | 5.9E-8 | 4E-11 | 4.5E-12 | .00026 | 5.9E-6 |
| | | Hazard Ratio (HR) | .74 | .69 | .68 | .74 | .74 |
| | p53 Mutated | p value | .0056 | .41 | .82 | .27 | .015 |
| | | Hazard Ratio (HR) | 1.99 | .82 | .95 | 1.5 | 2.29 |
| Ovarian[*] | p53 Mutated | p value | .00039 | .039 | .1 | .021 | .00075 |
| | | Hazard Ratio (HR) | 1.55 | .79 | .83 | 1.51 | 1.74 |
| Lung | All | p value | .00029 | 3.4E-7 | 1.4E-6 | 4.4E-5 | .18 |
| | | Hazard Ratio (HR) | 1.26 | .65 | .73 | 1.49 | 1.12 |
| | Adenocarcinoma | p value | 5.6E-9 | 5.5E-10 | 3.1E-7 | 1.7E-5 | 1.6E-8 |
| | | Hazard Ratio (HR) | 2.01 | .46 | .53 | 2.49 | 1.98 |
| Kidney | Renal Clear Cell | p value | 7.1E-06 | 2.1E-7 | 3.3E-05 | .2 | 9.0E-06 |
| | | Hazard Ratio (HR) | 1.98 | .46 | .53 | 1.42 | 2.75 |
| Type | Subtype | Variable | *INHA* | *ENG* | *INHA and ENG* | *INHA in High ENG Patients* | *INHA in Low ENG Patients* |
| Breast[#] | All | p value | 5.9E-8 | .0014 | 7.4E-6 | .0043 | .0027 |
| | | Hazard Ratio (HR) | .74 | .84 | .78 | .79 | .79 |
| | p53 Mutated | p value | .0056 | .12 | .057 | .26 | .035 |
| | | Hazard Ratio (HR) | 1.99 | 1.46 | 1.6 | .69 | 2.24 |
| Ovarian[*] | p53 Mutated | p value | .00039 | .0098 | .00091 | 1.8E-6 | .18 |
| | | Hazard Ratio (HR) | 1.55 | 1.36 | 1.49 | 2.12 | .8 |
| Lung | All | p value | .00029 | .0056 | .063 | .47 | 5.8E-8 |
| | | Hazard Ratio (HR) | 1.26 | 1.2 | 1.13 | 1.07 | 1.66 |
| | Adenocarcinoma | p value | 5.6E-9 | 1.6E-8 | 5.6E-12 | .14 | .00041 |
| | | Hazard Ratio (HR) | 2.01 | 1.98 | 2.3 | 1.25 | 2.12 |
| Kidney | Renal Clear Cell | p value | 7.1E-06 | 8.6E-6 | 4.5E-05 | .072 | 2.5E-06 |
| | | Hazard Ratio (HR) | 1.98 | .51 | .53 | 1.53 | 2.6 |

Survival curves were generated in KM Plotter for all cancer types. Survival curves represent overall survival, progression free survival (marked with [*]), or relapse free survival (marked with [#]) for patients expressing high or low mRNA (split by median) of the indicated gene.

adenocarcinomas patients expressing low *ENG* (HR = 2.12, p = 0.00041) but was not significant in *ENG* high expressing patients (HR = 1.25, p = 0.14) (Table 1).

Together, these findings suggest that *INHA* expression as a predictive tool for survival is influenced by the coreceptors *ENG* and *TGFBR3* in renal clear cell, lung, and p53 mutated breast and ovarian cancers. *INHA* is dependent on these coreceptors in all breast and ovarian cancers.

## Inhibins and activins can predict response to chemotherapy in luminal A breast cancer

We next evaluated the pathological response based classification for each of the genes using the receiver operating characteristic (ROC) plotter (www.rocplot.org) to validate and rank *INHA*, *INHBA/B*, *TGFBR3* and *ENG* as predictive biomarker candidates [110]. In a ROC analysis, an area under the curve (AUC) value of 1 is a perfect biomarker and an AUC of 0.5 corresponds to no correlation at all. We first entered all genes to allow for FDR calculation for each gene at FDR cutoff of 5% (S3 Table). We next examined individual genes and find that in luminal A breast cancers *ENG*, *TGFBR3*, *INHA*, and *INHBA*, were better performing as compared

to *INHBB* particularly for taxane or anthracycline based chemotherapy regimens. ROC plots for the two regimens are displayed in Fig 4 and S3 Table.

Both *ENG* and *TGFBR3* were predictive in other cancer types as well (S3 Table). Specifically, while *ENG* performed better in taxane treatments in *HER2*⁺ breast cancer subtype, *TGFBR3* performed better for taxane regimens in triple-negative breast cancer (TNBC) and serous ovarian cancer. Interestingly, examining expression (Fig 4b) revealed that in the same luminal A breast cancers *INHA*, *ENG* and *INHBA* are less expressed in responders to pharmacological treatment while *TGFBR3* is more expressed in these responder groups (Fig 4b). Similar trends for *TGFBFR3* expression were seen in TNBC and serous ovarian cancer groups where *TGFBR3* was more expressed in the responders' group for taxane regimens. *ENG* was also more expressed in *HER2*⁺ breast cancer patients who respond to taxane therapy, which was opposite to the luminal A subtype expression levels (Fig 4b). Full data for the ROC curve assessment is available in S3 Table. In summary *INHA*, *INHBA*, *TGFBR3*, and *ENG* display clear discrepant profiles of expression among responders and non-responders to both anthracycline and taxane chemotherapy for distinct breast cancer subtypes, specifically luminal A, and serous ovarian cancer. These genes also harbor a possible predictive value to indicate responsiveness to these therapy regimens. Moreover, *ENG* expression could also differentiate luminal A and *HER2*⁺ breast cancers response to taxane therapy. *INHBB* on the other hand had no predictive value in the assessed cancer types.

## Gene signatures from inhibins can predict patient survival outcomes

*INHA*, *TGFBR3*, and *ENG* impact patient outcomes more broadly and more significantly that *INHBB* and *INHBA*. There is also direct functional dependency of *TGFBR3* and *ENG* to inhibin rather than activin [38, 132]. We thus examined signatures associated with either a negative or positive outcome for each of the three genes. Cancer types that presented different survival predictions for *INHA*, *TGFBR3*, or *ENG* were assessed (Fig 5a), and cancer types in which each gene would have a similar patient outcome (i.e., positive overall survival outcome *vs*. negative overall survival outcome) were separated into groups (e.g., *INHA* positive outcome *vs*. negative outcome, Fig 5a).

Spearman's ρ coefficient was calculated for all RNA-seq gene data provided in each of these datasets, and values were clustered, and genes that were either positively and negatively correlated with each individual *INHA*, *TGFBR3* or *ENG* genes were identified (S4 Table). The top correlated genes from the positive outcome set were then pairwise compared to genes that had lower correlations in the negative outcome set, and vice-versa to obtain a subset of common genes [133–136]. Examples include *TGFB2* and *HOXA1* where genes correlated to *INHA* in the negative outcome set, and *OGG1* and *STAP2* in the positive outcome group. For *TGFBR3*, *AP1M1* and *RILPL1* correlated in the negative outcome context, while *FZD5* and *MYCN* in the positive one. No gene signatures were obtained for *ENG*. As indicated in section 3.2, the HR value range was the smallest for *ENG* in the assessed cancer types, which limits the differential gene signature analysis. All these genes also had their mRNA expression assessed in the respective cancer sets, contrasted, and evaluated for difference in expression (Fig 5b). Except for 22 genes from sets in which *INHA* or *TGFBR3* had distinct predictions of survival (e.g., *CHSY1*, *LDLR*, *PPARG*, *MIA2*, *TOX3*) all others exhibited significant alterations in gene expression (Fig 5b).

The altered genes from Fig 5b whose difference in expression was significant, were assessed for protein interactions and these iterated for pathway analysis using BioGRID (thebiogrid. org, Fig 5c) [111]. We find that *INHA* gene sets were associated with either *PD-L1* expression and PD-1 checkpoint, Rap1 signaling pathways in patients with positive outcomes or cell cycle

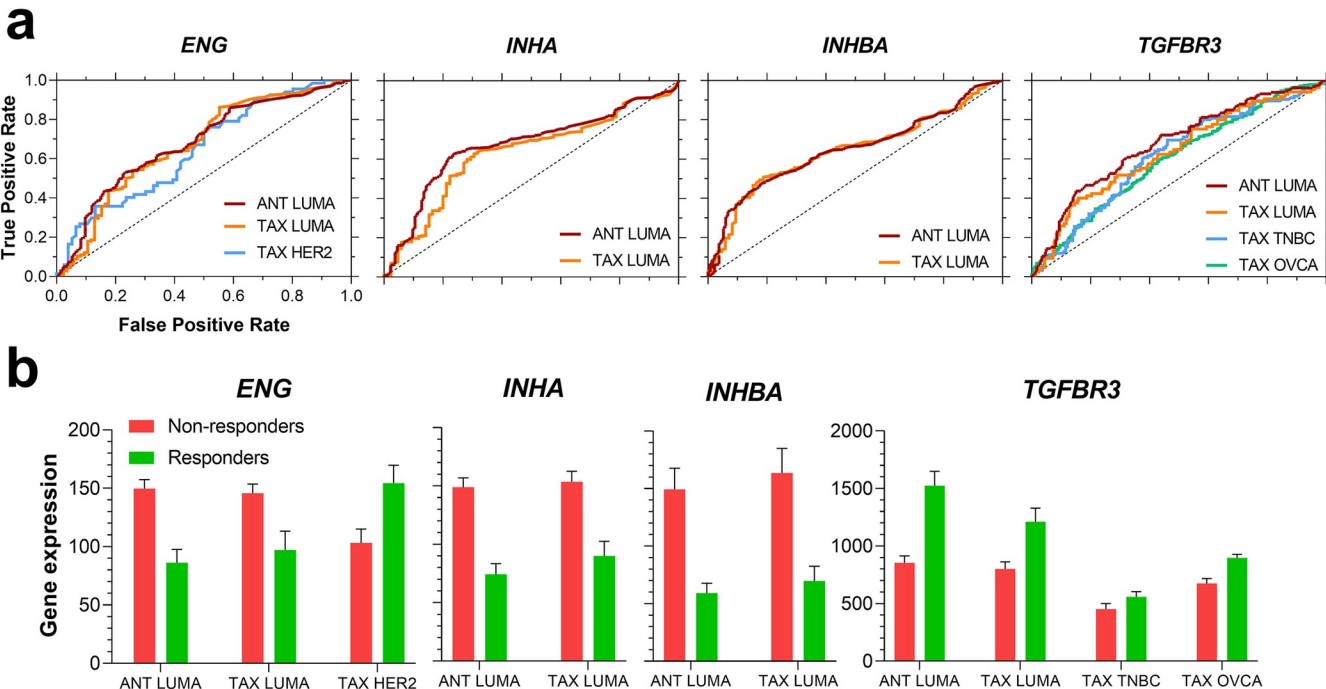

**Fig 4. ROC plots (a) and gene expression (b) of indicated genes for different chemotherapy regimens.** (a) ROC curves, in which performance ability was verified (i.e., AUC > 0.6), were plotted for *ENG*, *INHA*, *INHBA*, and *TGFBR3*. (b) Gene expression for each investigated gene between responders and non-responders for the assessed pharmacological treatments. The sample sizes for each group were the following: ANT LUMA, n = 474; TAX LUMA, n = 375; TAX HER2, n = 143; TAX TNBC, n = 290; TAX OVCA, n = 851. Abbreviations: ANT: anthracycline; TAX: taxane; LUMA: luminal A; TNBC: triple-negative breast cancer; OVCA: serous ovarian cancer.

regulation in patients with negative outcomes. *TGFBR3* associated genes on the other hand, relied on *VEGF* and *MAPK* signaling pathways for patients with positive outcome and *IL-17*, *p53*, or even *Wnt* signaling pathways in the negative outcome scenario. Detailed descriptions of analyzed genes and pathways are compiled in S4 Table.

To determine if the genes associated with *INHA* and *TGFBR3* had true prognostic value, a Probit regression model was applied to the normalized mRNA expression of the genes in S5 Table. The regressions were analyzed for the cancers from Fig 2a and 2d which had clear outcomes for either *INHA*, or *TGBFR3*. The final coefficients and entry genes are also provided in S5 Table. We find that the *INHA* model had 43 genes as dependent elements, and the *TGFBR3* model had 37 genes. However, the most suitable model obtained from these sets is the *TGFBR3* model, which has a high goodness of fit p-value (p = 0.9494), sensitivity (98.42%), specificity (91.56%), and accuracy (96.70%, Table 2).

These analyses reveal that a differential signature obtained from *INHA*, along with one of its main binding receptors (i.e., *TGBFR3*) are able to faithfully predict a patient's outcomes in a wide spectrum of cancer types (e.g., kidney, lung, head and neck, breast, liver, ovarian, stomach, endometrial).

## Functional analysis and interpretation of inhibin's mechanism of action

Prior functional studies indicate a dependency on *ENG* and *TGFBR3* for inhibin responsiveness [38, 132]. To test if these biological observations hold in patient datasets, we performed supervised clustering using Euclidean algorithm of genes correlating with either *INHA*, *ENG* or *TGFBR3* using the RNA-seq data for cancer types with the most significant impact as

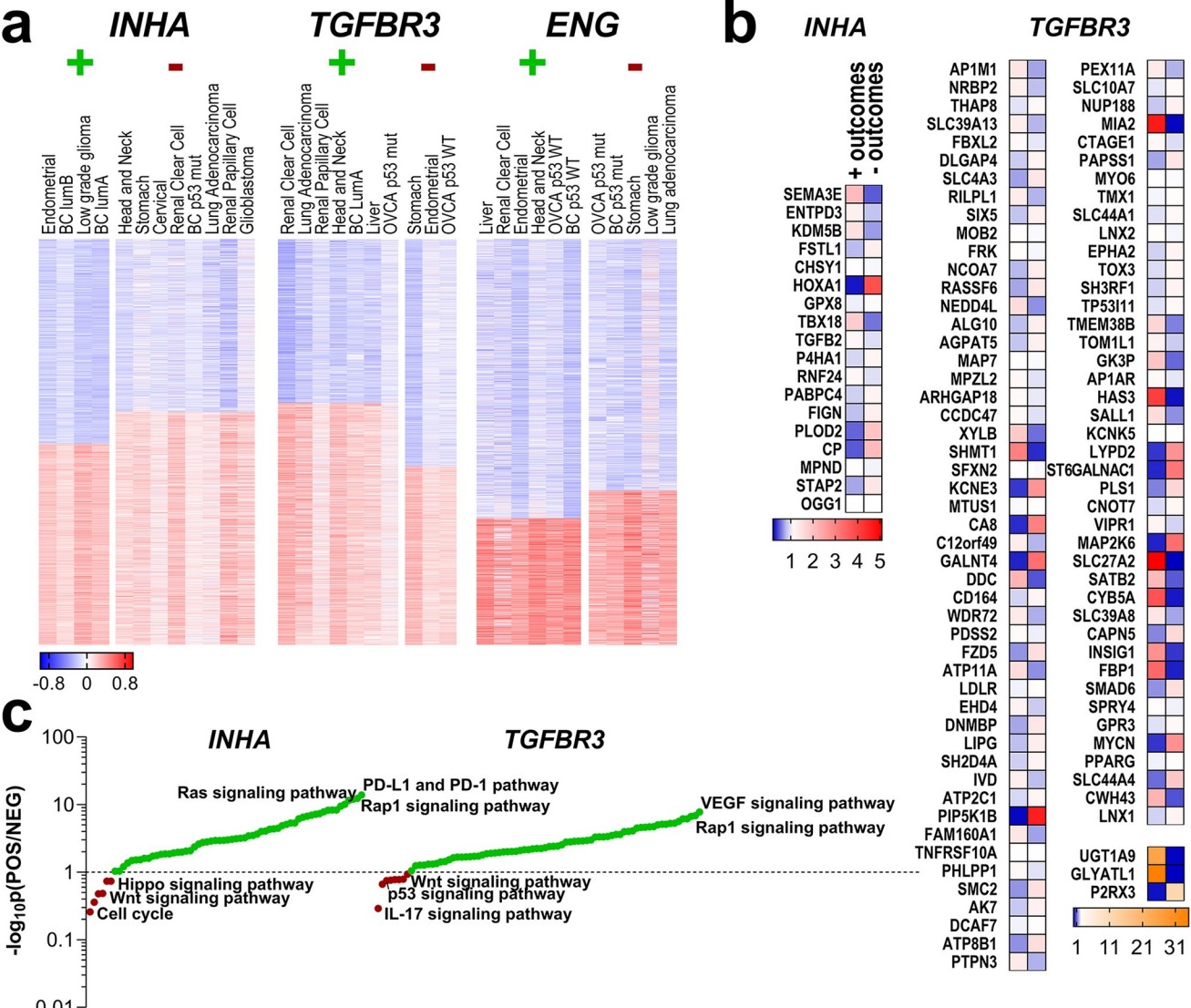

**Fig 5. Gene signatures and expression patterns for cancers where _INHA_, _TGFBR3_, or _ENG_ predicted survival outcomes.** (a) Cancer types in which either _INHA TGFBR3_ or _ENG_ had either positive (+) or negative (-) survival outcomes had their RNA-seq gene data correlation clustered for either low or high degree of correlations to each _INHA, TGFBR3_ or _ENG_ as indicated. (b) mRNA abundance of a subset of common genes obtained from pairwise comparisons of the top correlated genes from the positive outcome with the genes that had lower correlations in the negative outcome set, and vice-versa. mRNA expression was assessed in each cancer set.* p < 0.05 ** p < 0.01 *** p < 0.001 **** p < 0.0001 (c) Pathway analysis after BioGRID assessment of the significant genes from (b), ranked with a ratio of significance between sets from the positive and negative outcomes for each gene.

determined in Fig 6a. Only the most enriched transcripts that were either positively or negatively correlated transcripts are shown in Fig 6a. Most enriched genes from these clusters were then compared amongst each other in all pairwise combinations for similarities (e.g., positively correlated to _INHA vs._ negatively correlated to _TGFBR3_, and so on, Fig 6b).

We find that _INHA_ and _TGFBR3_ comparison rendered 1,430 genes, in which 24.6% were exclusive to _INHA_ (e.g., _DLL3, GPC2, TAZ, TERT, XYLT2_) 37.7% to TGFBR3 (e.g., _CCL2, CCR4, EGFR, GLCE, IL10RA, IL7R, ITGA1, ITGA2, JAK1, JAK2, SRGN, SULF1, TGFBR2_), and 13.1% were positively correlated to both (e.g., _CSPG4, COL4A3, FGF18, NOTCH4, SMAD9_). When _INHA_ was assessed with respect to _ENG_ we find 1,773 genes of which, 11.2% were

**Table 2. Prognostic performance of each delineated probit model.**

| | *INHA* model | *TGFBR3* model |
|---|---|---|
| Cancer types (+) | • Endometrial;<br>• BC lum A;<br>• BC lum B;<br>• Low grade glioma | • Renal Clear Cell;<br>• Lung adenocarcinoma;<br>• Renal Papillary Cell;<br>• Head and Neck;<br>• BC lum A;<br>• Liver;<br>• OVCA p53 mut |
| Cancer types (-) | • Head and Neck;<br>• Stomach;<br>• Cervical;<br>• Renal Clear Cell;<br>• BC p53 mut;<br>• Lung adenocarcinoma;<br>• Renal Papillary Cell;<br>• Glioblastoma. | • Stomach;<br>• Endometrial;<br>• OVCA p53 WT. |
| Genes in model | 43 | 37 |
| Specificity | 90.76% | 98.42% |
| Sensitivity | 93.17% | 91.56% |
| False positives ratio | 6.83% | 8.44% |
| False negative ratio | 9.24% | 1.58% |
| Accuracy | **92.25%** | **96.70%** |

For either an *INHA or TGFBR3* model, the described cancer types and subtypes used to analyze the positive (+), and negative (-) outcomes are shown. The number of genes in each model, the model's specificity (i.e., how the model certifies true positives), sensitivity (i.e., how the model certifies true negatives), and their false positive and false negative ratios are shown. The correct classification ratio is also highlighted below.

exclusive to *INHA* (e.g., *GDF9, PVT1*), 21.3% to *ENG* (e.g., *CCL2, GPC6, IL10, IL10RA, IL7R, INHBA, ITGA1, ITGB2, JAK1, SRGN, SULF1, TGFB1, TGFBR2*) and 10.0% were highly correlated to both (e.g., *CSPG4, DLL1, FGF18, FZD2, NOTCH4*). Lastly, the comparison between *ENG* and *TGFBR3* returned 1,938 genes. However, very few were exclusive to either *TGFBR3* (2.84%) or *ENG* (0.16%), revealing a high functional resemblance between both of these receptors, as most of the profiled genes correlated to both of them (48.5%, e.g., *ADAM9, -23, ADAMTS1, -2, -5, -8, -9, CCL2, CSF1R, DLL4, ESR1, FGF1, FGF2, FGF18, GLI1, -2, -3, GPC6, IL10RA, ITGA1, ITGA5, JAK1, MMP2, SDC3, SRGN, SULF1, TGFB3, TGFBR2, TNC, TWIST2, XYLT1, ZEB1*) or none of them.

We next used each gene set from the cross-comparisons in Fig 6b to identify pathways using KEGG [137]. Unique pathways with an FDR below 5% were identified for the comparisons and are presented in Fig 6c. Although several common pathways were present between groups, such as PI3K-Akt and Ras signaling pathways (see S6 Table), some unique pathways were present as well. *ENG*, for instance, was more exclusively related to cytokine-cytokine receptor interaction and natural killer mediated cytotoxicity (Fig 6c), while *TGFBR3* was more exclusively related to proteoglycans interaction and chemokine signaling. While cell cycle and DNA replication were not directly associated with *ENG* and *TGFBR3*, Rap1 signaling and Extracellular matrix (ECM)-receptor interactions were both impacted by *ENG* and *TGFBR3* (Fig 6c). However, no independent pathway could be pinpointed to *INHA* alone, revealing dependency on either *TGFBR3* or *ENG*. These studies indicate that inhibin's effects may vary

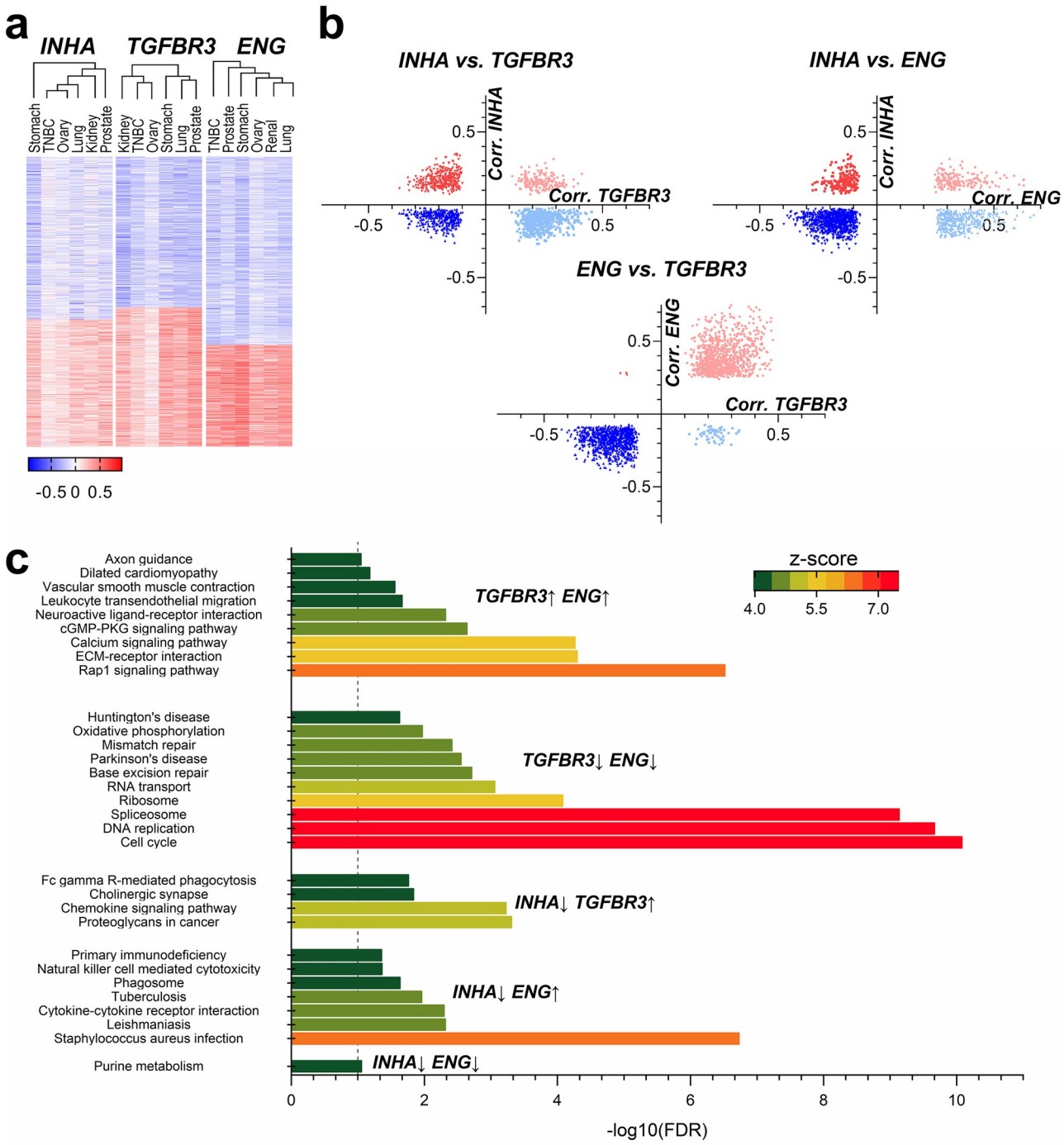

**Fig 6. Functional analysis of gene signatures between *INHA* and *TGFBR3* and *INHA* and *ENG*.** (a) Supervised clustering of correlations of RNA-seq data between *INHA*, *TGFBR3*, and *ENG* was performed to obtain sets of positive and negatively correlated genes for each set. (b) Common genes that were found in each group of correlated genes (e.g., negative correlation to *INHA* vs. positive to *TGFBR3* and all combinations) is presented. (c) KEGG pathway analysis for groups of genes correlated with the indicated combination. Unique pathways with an FDR below 0.05 were identified for the comparisons and are presented.

depending on whether *ENG* is more highly expressed as compared to *TGFBR3* with significant relevance to defining mechanism and impact of changes in components of this pathway.

## Discussion

This study aimed to evaluate comprehensively the influence of the inhibin-activin network in cancer. Our findings provide significant new information on the specific cancers impacted by the genes investigated here, *INHA*, *INHBA*, *INHBB*, *ENG* and *TGFFBR3*, and shed light on potential functional dependencies. Additional gene signature analysis reveals that *INHA*, along with one of its main receptors (i.e., *TGBFR3*) faithfully predicts patient outcomes in a wide spectrum of cancer types.

TGFβ-1 is a representative member of the TGF-β family that has been significantly investigated previously [138]. However, less information exists about the precise impact and role of other members like inhibins and activins. Our findings that *INHA* is significantly associated with survival in sixteen of the twenty cancers analyzed, correlating positively with survival in five cancers and negatively in ten (Fig 2), highlight *INHA*'s differential role as a tumor suppressor or promoter depending on the specific cancer type. In highly angiogenic tumors like renal clear cell carcinoma [139] and glioblastoma [140], we found *INHA* expression to be a significant negative predictor of survival. *INHA*'s role in promoting tumorigenesis in these cancer types may occur through its effects on angiogenesis as has been previously reported for a subset of ovarian and prostate cancer [27, 72] warranting further investigation. In Luminal A breast cancers, we observed that increased *INHA* expression was associated with unresponsiveness to chemotherapy (Fig 4) while in survival data it was a positive predictor of survival (Fig 2). This apparent contradiction can perhaps be explained by the fact that data in KM Plotter contains information on patients that have undergone a wide array of treatments. Likely, *INHA* is predictive of response to some treatments but not others. In both breast and ovarian cancers, *INHA* was a negative predictor of survival in patients that had p53 mutations indicating a potential dependency of *INHA* functions on the p53 status. *INHA* expression alterations have been observed in p53 mutated adrenocortical tumors and *INHA* was suggested to be a contributing factor to tumorigenesis in these cancers [141]. One of the most characterized transcriptional activators of *INHA* is GATA4 [142], which can also regulate p53 in cancer and could contribute to the different survival outcomes observed for *INHA* in p53 mutated cancers versus wild-type p53 cancers [143, 144]. *INHA's* link to functional outcomes in the background on p53 mutations remains to be fully elucidated.

Between the TGF-β family co-receptors (*ENG* and *TGFBR3)* implicated in cancer progression and inhibin function, *ENG* was more expressed (Fig 1c), particularly in lung adenocarcinoma and gastric cancers, corresponding with *ENG* being a strong negative predictor of survival (Figs 1c and 2). These findings are consistent with prior experimental findings as well [145, 146]. In p53 mutated cancers, *ENG* remained a negative predictor. ROC Plotter analysis revealed decreased *ENG* expression to be associated with response to anthracycline therapy in Luminal A breast cancer patients (Fig 4). However, a previous study showed that positive *ENG* expression was associated with increased survival in breast cancer patients who had undergone anthracycline treatment [147]. While Isacke and colleagues did not report a specific subtype in their analyzed cohort [147], we obtained significant results for Luminal A breast cancer, specifically. Moreover, an additional study performed in acute myeloid leukemia showed an inverse relationship to that of Isacke et al., consistent with our results in Luminal A breast cancer [147, 148]. We also found *ENG* to be a predictive of response to taxane therapy regimens. An inverse relationship between *ENG* expression was observed in responders for Luminal A and *HER2*[+] breast cancer, with responders expressing high *ENG* in *HER2*[+] breast cancers but low levels of

*ENG* in Luminal A cancers (Fig 4). As Luminal A breast cancer is *HER2*⁻, *ENG* could be affected by HER2 status in these cancer types. In our analysis, expression data was only obtained for Luminal A breast cancers not *HER2*⁺ so differences in expression between the two were not analyzed.

Consistent with *TGFBR3*'s role as a tumor suppressor in many cancers, we found it to be a significant positive predictor of survival in all but two cancers (i.e., endometrial and all gastric subtypes, Fig 2). Increased *TGFBR3* was predictive of response in all treatments and cancers we examined (Fig 4), further bolstering *TGFBR3*'s role as a negative regulator of tumor progression. Specifically, Bhola et al. (2013) [149] showed increased levels of *TGFBR3* in response to taxane in a small cohort (n = 17) of breast cancer patients; however, response to therapy was not analyzed. *TGFBR3* has been shown to act as a tumor suppressor in renal clear cell carcinoma [127] and non-small cell lung cancers [102] which was also confirmed here (Fig 2). We were also able to expand *TGFBR3*'s role in renal cancer to papillary carcinomas as well (Fig 2).

Expression of *ENG* and *TGFBR3* was not significantly different between wild-type and p53 mutated cancers indicating p53 likely does not impact expression itself. Whether protein secretion of these coreceptors is altered in these cancers is currently unknown, and cannot be ruled out, as previous studies have shown increased endoglin folding and maturation in p53 mutation settings [150]. TGFBR3 also undergoes N-linked glycosylation, so a similar scenario to endoglin is possible. Alterations in protein maturation could explain the differential patient outcomes observed between wild-type and p53 mutated cancers, when assessing for *ENG* and *TGFBR3*, despite changes in expression not being observed.

*INHA*'s dependency on each coreceptor examined in survival analysis revealed distinct signatures between different cancer types (Table 1). Prior studies indicate a requirement for *ENG* in inhibin responsiveness and functions [27], which was borne out in patient survival data here (Table 1). However, a few outliers exist such as p53 mutated breast and renal clear cell carcinoma where *INHA* was not always dependent on increased *ENG* and *TGFBR3* expression. We found *INHA* to only be a negative predictor of survival in patients expressing low *ENG* indicating *INHA* might act independent of either coreceptor in these cancer types. The role of other receptors involved in mediating *INHA*'s effects in these cancer types remains to be determined.

Betaglycan and endoglin are co-receptors for TGFβ-1,2,3 and have been shown to regulate signaling for isoforms of BMP, Wnt and FGF [151–153]. However, both endoglin and betaglycan are dispensable for response to the above-mentioned growth factors, playing primarily modulatory functions. Given that TGF-β's BMPs, Wnt, and FGF can act as both tumor suppressors and promoters in a cancer and context dependent manner, and our analysis indicating that *ENG* and *TGFBR3* are both strong predictors of survival on their own (Fig 2) it is likely that *ENG* and *TGFBR3* expression levels impact signaling sensitivity and thereby patient outcomes in the context of those signaling ligands.

In contrast to the above listed growth factors, Inhibins are reported to have functional consequences that dependent primarily on betaglycan or endoglin [23, 27] consistent with the ability of the gene signatures (Fig 5) dependent on *TGFBR3* and *ENG* to distinguish patients' outcomes. Some notable elements of this signature have been verified previously and even proposed as cancer biomarkers. For example, *EPHA2* overexpression has been associated with decreased patient survival and promotes drug resistance, increased invasion, and epithelial to mesenchymal transition (EMT) [154–157]. HOXA1, a lncRNA overexpressed in cancers such as breast, melanoma, and oral carcinomas, drives metastasis and tamoxifen resistance [158–160]. For *TGFBR3* specifically, three genes revealed high discrimination between positive and negative outcomes: *UGT1A9* and *GLYATL1* were 25- and 35-fold more expressed in positive outcomes and *P2RX3* was 11.5-fold more expressed in negative outcomes. Of interest,

UGT1A9 is a UDP-glucuronosyltransferase (UGT) whose activity has been implicated in drug resistance by affecting the bioactivity of the drug [161, 162]. We speculate that as a proteogly-can, increased *TGFBR3* could compete for UDP-glucuronate acid (GlcA) and UDP-xylose, both key elements for UGT1A9 activity, thereby potentially disrupting UGT associated resistance mechanisms and increasing the efficacy of chemotherapy. We also narrowed down which pathways differentiated patient outcomes for either *INHA* or *TGFBR3*. For positive outcomes, we found that *INHA* was associated with PD-L1, Ras, and Rap1 signaling pathways. In adverse outcomes, *INHA* was associated with Hippo, Wnt, and cell cycle pathways. Wnt has been shown to regulate *INHA* transcription in rat adrenal cortex and could increase *INHA* expression in certain tumors to promote tumorigenesis [163]. Recent evidence indicates increased PD-L1 in dendritic cells in INHA$^{-/-}$ mice [164]. We speculate that increased *INHA* in tumors may inhibit PD-L1 expression perhaps via antagonistic effect on other TGF-β members, increasing anti-tumor immune responses.

There are currently no other cancer prognostic models based on our three assessed genes. The selected prognostic model showed high accuracy (96.7%) with 98.42% sensitivity and 91.56% specificity (Table 2). Moreover, most prognostic cancer models are directed to either a specific cancer type (e.g., breast, prostate) or a cancer stage (e.g., lymph node metastases, phases). Our model includes at least ten tumor types, is in the top two for sensitivity, and among the second quartile of specificities on assessment of 48 prognostic cancer models [165–168]. Thus, the *INHA-TGFBR3-ENG* signature has pan-cancer prognostic value. Interestingly, there were very few SMAD and canonical TGF-β associated pathway members that were part of the probit analysis (S5 Table). However, several genes associated with non-SMAD TGF-β signaling were included, such as *MAP2K6*, *FZD5*, and *PHLPP1* which are associated with MAPK, Wnt, and Akt signaling pathways respectively [169]. Much of TGF-β's functions in EMT, invasion and metastasis have been associated with non-SMAD pathways [169, 170] which are more likely to involve the coreceptors *TGFBR3* and *ENG*. Hence it was not surprising that such non SMAD pathways were predominant in the *INHA-TGFBR3-ENG* analysis.

Clustering analysis for genes correlated with *INHA*, *TGFBR3*, or *ENG* in cancers (Fig 6) revealed *ENG* and *TGFBR3* had very few genes correlated exclusively to one or the other. As both receptors share similar structures and interact with common ligands [38], this is not unexpected. Similarly, since *ENG* and *TGFBR3* had significant common gene associations this resulted in common pathways. For instance, a strong correlation with ECM-receptors and Rap1 signaling was observed. ENG has been shown to bind leukocyte integrins, promoting invasion [171], and ECM remodeling during fibrosis [172]. TGFBR3 has been shown to regulate integrin localization and adhesion to ECM [173]. *ENG* alone was associated with natural killer cell-mediated cytotoxicity consistent with previous findings showing anti-endoglin therapy augmented immune response in tumors by increasing NK cells, CD4$^+$, and CD8$^+$ T lymphocytes [174].

In conclusion, our pan-cancer analysis of the inhibin-activin network reveals a prognostic signature capable of accurately predicting patient outcome. Gene signatures from our analysis reveal robust relationships between *INHA*, *ENG*, and *TGFBR3* and other established cancer biomarkers. Survival analysis implicated members of the inhibin-activin network in cancers previously unstudied as well as corroborated previous findings. Further analysis of the role of the inhibin-activin network in cancer and relationship to other cancer associated genes, as well as validation as predictive biomarkers to chemotherapy is needed.

## Supporting information

**S1 Table. Supplementary data regarding gene expression, alteration and dependency.**
(XLSX)

**S2 Table. Complementary table for survival data.**
(DOCX)

**S3 Table. Full data concerning ROC analysis.**
(XLSX)

**S4 Table. INHA, TGFBR3 and ENG survival gene signature full data discrimination.**
(XLSB)

**S5 Table. Probit regression detailed outputs.**
(XLSX)

**S6 Table. INHA-TGFBR3-ENG pairwise gene comparison data.**
(XLSX)

## Author Contributions

**Conceptualization:** Eduardo Listik, Ben Horst, Nam. Y. Lee, Karthikeyan Mythreye.

**Formal analysis:** Eduardo Listik, Ben Horst, Alex Seok Choi, Karthikeyan Mythreye.

**Funding acquisition:** Karthikeyan Mythreye.

**Investigation:** Ben Horst, Karthikeyan Mythreye.

**Methodology:** Eduardo Listik, Ben Horst, Balázs Győrffy, Karthikeyan Mythreye.

**Supervision:** Karthikeyan Mythreye.

**Validation:** Eduardo Listik, Balázs Győrffy.

**Visualization:** Eduardo Listik, Ben Horst, Alex Seok Choi, Karthikeyan Mythreye.

**Writing – original draft:** Eduardo Listik, Ben Horst, Karthikeyan Mythreye.

**Writing – review & editing:** Eduardo Listik, Ben Horst, Alex Seok Choi, Nam. Y. Lee, Balázs Győrffy, Karthikeyan Mythreye.

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
