## [Decision Letter · Decision Letter 0]

29 Jan 2021

PONE-D-20-36221

A bioinformatic analysis of the inhibin-betaglycan-endoglin/CD105 network reveals prognostic value in multiple solid tumors

PLOS ONE

Dear Dr. Mythreye,

Thank you for submitting your manuscript to PLOS ONE. After careful consideration, we feel that it has merit but does not fully meet PLOS ONE’s publication criteria as it currently stands. Therefore, we invite you to submit a revised version of the manuscript that addresses the points raised during the review process. Specifically, please expand the Discussion to address the two points raised by Reviewer #1 and improve the image resolution as mentioned by Reviewer #2.

We look forward to receiving your revised manuscript.

Kind regards,

Donald P. Bottaro

Academic Editor

PLOS ONE

Journal Requirements:

2.We note that the grant information you provided in the ‘Funding Information’ and ‘Financial Disclosure’ sections do not match.

3. Please ensure that you refer to Figure 3 in your text as, if accepted, production will need this reference to link the reader to the figure.

Reviewers' comments:

Reviewer's Responses to Questions

**Comments to the Author**

1. Is the manuscript technically sound, and do the data support the conclusions?

Reviewer #1: Yes

Reviewer #2: Yes

2. Has the statistical analysis been performed appropriately and rigorously? 

Reviewer #1: Yes

Reviewer #2: Yes

3. Have the authors made all data underlying the findings in their manuscript fully available?

Reviewer #1: Yes

Reviewer #2: Yes

4. Is the manuscript presented in an intelligible fashion and written in standard English?

Reviewer #1: Yes

Reviewer #2: Yes

5. Review Comments to the Author

Reviewer #1: The manuscript from Listik and colleagues is a bioinformatics analysis to evaluate the significance of a specific signaling axis - inhibin signaling axis- in different cancer types. The choice of the signaling axis is justified from published studies as cited by the authors. Publicly available cancer patient databases have been used and analysis has been well described in the methods. Overall, these findings point to a bioinformatic based mechanistic link between the proteins examined. Additionally the predictive outcome on different cancers is highly useful, as these proteins and this axis have thus far not been systematically ever examined and are likely to create avenues for new investigation. The identification of a gene signature that could potentially predict patient’s outcomes, is highly interesting as well

Minor points to be addressed

TGFBR3 or betaglycan, is a coreceptor for all three TGF-β isoforms, especially for TGF-β2, and heterodimer inhibin A (InhA) BMPs and GDFs, Wnt/b-catenin signaling, as reported by the authors.  Since TGFBR3 has predictive value in cancer patients, authors should include in their discussion the potential impact of TGFBR3 changes on other pathways in discussion and similarly for endoglin CD105.

Regarding signature genes:

In Table S5, few known SMAD2/3 target genes were in the list suggesting non SMAD mechanisms downstream of this axis.  Authors should discuss this as well.

Considering the broad scope of the journal and its readers, at few places, the text may be further simplified and sentences shortened.

Reviewer #2: The presented manuscript, Listic et al., aims to determine the inhibins and betoglycan/endoglycan network's predictive value on tumor progression and the diseases' outcome by analyzing public databases. Authors utilized cBiPortal, DepMap, ROC Plotter, and KM Plotter. The combinatorial analysis revealed a robust relationship between the expression of INHA, ENG, and TGFBR3 and the progressions of cancers of multiple origins.

The manuscript is convincing and performed at a high technical level. It is well written and easy to understand.

The only problem is the quality of the images. The current resolution is low, so it is difficult to read.

To conclude, the manuscript is suitable for publication in PLOS One after the correction of the quality of the images.

6. PLOS authors have the option to publish the peer review history of their article (what does this mean?). If published, this will include your full peer review and any attached files.

Reviewer #1: No

Reviewer #2: No

---

## [Author Response · Author response to Decision Letter 0]

17 Mar 2021

Response to Academic Editor and Reviewers:

• “Please ensure that your manuscript meets PLOS ONE's style requirements, including those for file naming.”

We do apologize for the divergence of style. We have thoroughly revised the documents, following both guidelines that were sent. We have edited:

o Headings and subheadings;

o Caption locations;

o In-text references of figures;

o Supplementary material text format;

o Title page;

o Figure and Supplementary material file names.

• “We note that the grant information you provided in the ‘Funding Information’ and ‘Financial Disclosure’ sections do not match.”

We regret the typo in the grant number. The sources of funding received for the research submitted to the journal are listed below. This information includes the name of granting agencies, grant numbers which have also been corrected in the manuscript data ' Funding section'.

National Cancer Institute

Award Number: R01CA219495 | Recipient: Karthikeyan Mythreye, Ph.D

National Research, Development and Innovation Office Hungary

Award Number: 2018-2.1.17-TET-KR-00001 | Recipient: Balázs Győrffy

National Research, Development and Innovation Office, Hungary 

Award Number: 2018-1.3.1-VKE-2018-00032 | Recipient: Balázs Győrffy

• “Please ensure that you refer to Figure 3 in your text as, if accepted, production will need this reference to link the reader to the figure.”

Our in-text references to the Figures have now been corrected /changed to “Fig 2 and 3” according to PloS manuscript formatting guidelines.

REVIEWER #1

MINOR POINTS

• “TGFBR3 or betaglycan.….Since TGFBR3 has predictive value in cancer patients, authors should include in their discussion the potential impact of TGFBR3 changes on other pathways in discussion and similarly for endoglin CD105.”

We appreciate the Reviewer’s point as both endoglin and betaglycan serve as co-receptors for other signaling pathways of the TGFβ superfamily and Wnt and FGF. Given the wide range of ligands we have now as suggested by the reviewer discussed the rationale and potential implications on these pathways in the broad context of carcinogenesis. These can be found on lines 509-518 in the final manuscript ( clean version).

• “In Table S5, few known SMAD2/3 target genes were in the list suggesting non SMAD mechanisms downstream of this axis. Authors should discuss this as well.”

We thank these reviewers for this point allowing us to highlight the significance of non-SMAD pathways in cancer. We have now added this to our discussion on lines 546-553 ( clean version)

• “Considering the broad scope of the journal and its readers, at few places, the text may be further simplified and sentences shortened.”

Changes have been made where appropriate.

REVIEWER #2

• “The only problem is the quality of the images. The current resolution is low, so it is difficult to read.”

We regret this and have now resubmitted higher resolution, 300-600 dpi, LZW-uncompressed, TIFF images which should offer better quality.

---

## [Editor Report · Decision Letter 1]

22 Mar 2021

A bioinformatic analysis of the inhibin-betaglycan-endoglin/CD105 network reveals prognostic value in multiple solid tumors

PONE-D-20-36221R1

Dear Dr. Mythreye,

We’re pleased to inform you that your manuscript has been judged scientifically suitable for publication and will be formally accepted for publication once it meets all outstanding technical requirements.

Kind regards,

Donald P. Bottaro

Academic Editor

PLOS ONE

---

## [Editor Report · Acceptance letter]

26 Mar 2021

PONE-D-20-36221R1 

A bioinformatic analysis of the inhibin-betaglycan-endoglin/CD105 network reveals prognostic value in multiple solid tumors 

Dear Dr. Mythreye:

I'm pleased to inform you that your manuscript has been deemed suitable for publication in PLOS ONE. Congratulations! Your manuscript is now with our production department. 

Kind regards, 

on behalf of

Dr. Donald P. Bottaro 

Academic Editor

PLOS ONE